# Synthetic NAC 71-82 Peptides Designed to Produce Fibrils with Different Protofilament Interface Contacts

**DOI:** 10.3390/ijms22179334

**Published:** 2021-08-28

**Authors:** Thomas Näsström, Tobias Dahlberg, Dmitry Malyshev, Jörgen Ådén, Per Ola Andersson, Magnus Andersson, Björn C. G. Karlsson

**Affiliations:** 1Physical Pharmacy Laboratory, Linnaeus University Centre for Biomaterials Chemistry, Linnaeus University, SE-392 31 Kalmar, Sweden; thomas.nasstrom@lnu.se; 2The Biophysics and Biophotonics Group, Department of Physics, Umeå University, SE-901 87 Umeå, Sweden; tobias.dahlberg@umu.se (T.D.); dmitry.malyshev@umu.se (D.M.); magnus.andersson@umu.se (M.A.); 3Department of Chemistry, Umeå University, SE-901 87 Umeå, Sweden; jorgen.aden@umu.se; 4Department of Material Science and Engineering, Applied Materials Science, Uppsala University, SE-751 03 Uppsala, Sweden; perola.andersson@angstrom.uu.se

**Keywords:** α-synuclein, NAC 71-82 peptides, fibril polymorphs

## Abstract

Alpha-synucleinopathies are featured by fibrillar inclusions in brain cells. Although α-synuclein fibrils display structural diversity, the origin of this diversity is not fully understood. We used molecular dynamics simulations to design synthetic peptides, based on the NAC 71-82 amino acid fragment of α-synuclein, that govern protofilament contacts and generation of twisted fibrillar polymorphs. Four peptides with structures based on either single or double fragments and capped or non-capped ends were selected for further analysis. We determined the fibrillar yield and the structures from these peptides found in the solution after fibrillisation using protein concentration determination assay and circular dichroism spectroscopy. In addition, we characterised secondary structures formed by individual fibrillar complexes using laser-tweezers Raman spectroscopy. Results suggest less mature fibrils, based on the lower relative β-sheet content for double- than single-fragment peptide fibrils. We confirmed this structural difference by TEM analysis which revealed, in addition to short protofibrils, more elongated, twisted and rod-like fibril structures in non-capped and capped double-fragment peptide systems, respectively. Finally, time-correlated single-photon counting demonstrated a difference in the Thioflavin T fluorescence lifetime profiles upon fibril binding. It could be proposed that this difference originated from morphological differences in the fibril samples. Altogether, these results highlight the potential of using peptide models for the generation of fibrils that share morphological features relevant for disease, e.g., twisted and rod-like polymorphs.

## 1. Introduction

Patients suffering from the neurodegenerative diseases Parkinson’s disease (PD), Lewy body dementia (LBD) and multiple system atrophy (MSA) display fibrillar inclusions of α-synuclein in neuronal and glial cells [1,2,3]. Although it is believed that the cell toxicity and associated pathology of a fibril are correlated to its structure, the whole picture is not fully understood—the role of different polymorphs and toxicity is still under debate. This knowledge gap has in turn limited the development of diagnostic methods that could be used to propose more efficient drug therapies for fibril-induced neurodegenerative diseases.

In attempts to develop better therapeutic strategies and diagnostic methods for α-synuclein-related disease, previous studies have been focused on mapping the correlation between aggregation propensity of certain parts of α-synuclein in nucleation and elongation of amyloid fibrils [4]. The normal physiological function of α-synuclein protein is believed to involve pre-synaptic vesicle transport and SNARE complex assembly [5]. Although α-synuclein in solution has been characterised as an intrinsically disordered protein, contact with surfaces and other macromolecules can induce it into folded conformations. For example, when α-synuclein binds to negatively charged lipid vesicles [6] and micelles, they fold into α-helical-rich conformations [7]. Furthermore, lowering the polarity of the solution by the presence of aliphatic alcohols can, in turn, induce β-sheet folds [8].

Structurally, α-synuclein is composed of 140 amino acids (aa) that are divided into three regions: an amphipathic N-terminal part (1–60 aa) which holds four 11-repeats (KTKEGV), a C-terminal part rich in acidic residues (96–140 aa) and a hydrophobic middle part (61–95 aa) known as the non-amyloid-β component (NAC) responsible for driving aggregation of full-length α-synuclein into β-sheet-rich oligomers and fibrils [9]. Furthermore, in vitro aggregation studies have revealed that by deleting the 71–82 aa stretch from the NAC region, aggregation of α-synuclein is prevented, thereby suggesting a crucial role of the 71–82 aa stretch for generation of fibrillar species of the full-length protein [10]. Structural characterisation of α-synuclein fibrils using cryo-EM revealed that protofilament contacts involving 61–83 aa (the 71–82 aa NAC region) [11] result in twisted polymorphs whereas contacts involving 37–61 aa yield rod-like structures [12,13].

Since the protofilament interface contacts are important for fibril polymorphism, we hypothesised that designed peptides made from aa that stabilise these contacts could generate fibrillar structures equivalent to wild-type fibrils. Such peptide-induced fibrillar structures would be interesting target motifs for developing sensor-based assays for fibril structure analysis in α-synucleinopathies. These peptides could have motifs representing the core of α-synuclein fibrils that is otherwise hard to target using conventional detection methods. Since fibrils can aggregate into complex forms and other proteins and biomolecules are present in a sample, it is challenging to classify aggregates using standard techniques. For example, X-ray crystallography, nuclear magnetic resonance spectroscopy and cryo-electron microscopy all require homogeneous samples. Therefore, Raman spectroscopic methods have previously been suggested useful for in situ structural characterisation of fibrils in biological fluid environments [14]. Although Raman spectroscopic techniques have successfully provided information on the extent of the secondary structure of fibrillar aggregates, this technique has so far reached limited success in detecting fibril polymorphs in crude fibrillar samples. To circumvent this limit, time-resolved fluorescence spectroscopy can be suggested to complement vibrational spectroscopic methods. TCSPC is a time-resolved fluorescence spectroscopic technique that can be applied to in situ measurements [15,16,17] and can potentially be used for detecting and characterising the structural diversity of fibrils in complex matrices.

Inspired by the importance of the NAC 71-82 aa and the potential of using peptide-induced fibrils for sensor development, we characterised the structural diversity of fibrils prepared from synthetic single- and double-fragment NAC 71-82 peptides in this work. Molecular dynamics simulations of the early oligomerisation stage of fibrillisation mixtures suggested the formation of more ordered aggregated forms for the studied non-capped double-fragment NAC 71-82 peptide compared to its capped counterpart. Generated fibrils were analysed by laser-tweezers Raman spectroscopy (LTRS), which suggested the formation of a more heterogeneous population of fibrillar structures for double-fragment peptides compared to the single-fragment peptides. Transmission electron microscopy (TEM) demonstrated, in addition to the presence of short protofibril structures in bundles in all fibril samples analysed, more elongated structures in double-fragment peptide fibrils. Elongated structures found in non-capped double-fragment fibrils were found to be twisted and the result from two intertwined protofilaments. In contrast, capped double-fragment fibrils were observed to be rod-like and built by single protofilaments. This structural stability of non-capped double-fragment peptides agreed with that predicted by molecular dynamics simulation data. Finally, we applied the TCSPC technique on fibrils incubated with the amyloid dye Thioflavin T (ThT). Results from these analyses suggested that this technique can be used to map the presence of mature fibrils in a complex fibrillar mixture. Altogether, the results presented in this work show the potential of using peptide models for the generation of fibrils that share morphological features relevant for disease, e.g., twisted and rod-like polymorphs.

## 2. Results and Discussion

### 2.1. Molecular Dynamics Simulations Suggest the Formation of Stable P2-L(G,G)-CT Peptide Aggregates

In the design of fibrils with a predicted variation in protofilament interface contacts, we used molecular dynamics (MD) simulations to characterise the underlying interactions governing peptide aggregation and nanofibre growth. See Appendix A, for snapshots of peptide mixtures before and after aggregation. We assessed the number and stability of peptide complexes formed during MD simulations. The population of aggregates and size formed for both peptide systems were similar: 4.2 ± 1.3 and 9.4 ± 2.0 for P2-L(G,G) and 4.2 ± 0.9 and 9.0 ± 2.5 for the P2-L(G,G)-CT peptide systems, respectively (Appendix A). These results suggest that the choice of N- and C-terminal charge model or solvent has little impact on the rate of the early oligomerisation stage of these peptide systems. In addition, calculation of the aggregation propensity (AP) from computed values of the decrease of the total solvent accessible surface area (SASA) for all peptides over time revealed similar distributions of AP for both peptide models, typically ranging between 1.4 and 1.9 (sampled values over time are shown in Appendix A). This is in line with that reported for lysine-containing tripeptides forming translucent self-supported hydrogels [18]. Moreover, AP values were calculated after excluding the possibility of peptide–peptide interaction, i.e., only including loss of SASA due to peptide folding. This calculation resulted in a maximum value of 1.3, indicating that ~23% of SASA for the peptides is lost due to folding. For peptide systems with the highest value of AP (~1.9), a total of ~47% decrease in SASA was calculated (with a decrease of ~24% due to the formation of peptide–peptide contacts).

Analysis of stable peptide aggregates formed during MD simulation (Appendix A) and calculation of aggregate shape descriptors (Appendix A) suggested the formation of a variety of aggregate sizes and shapes. Most of the aggregates were found to be amorphous, but more ordered aggregates were also observed (Figure 1).

The most stable peptide aggregate having a shape resembling a nanofibre was formed in the P2-L(G,G)-CT system after 90 ns of simulation (Appendix A). In this system, the aggregate was built up by 13 peptides with an acylindricity value close to zero and values for asphericity and relative shape anisotropy of ~0.5 and ~0.3, respectively. The packing of peptides in such a nanofibre assembly was found to be similar to that of a β-hairpin arrangement. The calculated AP value for this simulation replica was one of the lowest among the studied peptide mixtures (AP = ~1.4), indicating that the nanofibre is formed with a limited loss in SASA and only ~16% loss of SASA due to peptide contacts (total decrease of 33% and 17% due to peptide folding). Moreover, the elongated nanofibre shape of this peptide aggregate was also supported by computed ratios of the longest and shortest (L_X_/L_Z_) and the second-longest and -shortest (L_Y_/L_Z_) dimensions which were found to be ~5.0 and ~1.2, respectively. Notably, a smaller peptide nanofibre aggregate structure could also be observed in the P2-L(G,G)-CT system. A peptide aggregate (formed after 55 ns) was found to be less than half the size of the larger aggregate observed; however, yielded values in shape descriptors supported a nanofibre structure.

Aggregates formed in the P2-L(G,G) peptide system were in general smaller compared to those formed by P2-L(G,G)-CT peptides, and in addition to nanofibre structures, spherical aggregates were observed (see Figure 1). Calculation of the compactness (F-values, Appendix A) suggested that nanofibres demonstrated higher values of F and could thereby be associated with a less compact structure than that for other aggregate types.

To confirm the importance of the charged N- and C-termini of double-fragment NAC 71-82 peptide models for β-sheet layer stability, we calculated the distribution of the relative peptide end-to-end distances (R_T-T_). Calculated R_T-T_-values for all peptides in each simulation replica, as well as for all simulation replicas combined, (Appendix A) confirmed lower conformational flexibility for the P2-L(G,G)-CT compared to P2-L(G,G) peptide. This observation was based on a shift in calculated R_T-T_ distance population towards lower values for P2-L(G,G)-CT peptide aggregates.

Altogether this observation suggests that the P2-L(G,G)-CT peptide can more easily form a nanofibre structure than the P2-L(G,G) peptide when individual peptides are in an active state, that is, in a specific conformation. We believe that the higher conformational flexibility of the P2-L(G,G) peptide results in a decreased probability to be in a conformation that favours aggregation. This could lead to a destabilised planar β-hairpin conformation and thereby a destabilised β-sheet layer in the P2-L(G,G) protofilament structure. To investigate these observations and gain insight into peptide aggregation, we synthesised four types of fibrils and characterised their biophysical properties.

### 2.2. Synthesised P2-L(G,G)-CT Fibrils Demonstrated Structural Heterogeneity

Motivated by the results from MD simulations, we produced fibrils from single- and double-fragment NAC 71-82 peptides and analysed the fibrils using a variety of biophysical and biochemical techniques. Fibrils of single-fragment peptides (P1 and P1-CT) were prepared as controls. Interestingly, the growth of fibrils in the double-fragment NAC 71-82 peptide system (P2-L(G,G) and P2-L(G,G)-CT) was observed to be faster than for single-fragment NAC 71-82 peptides (P1 and P1-CT) (see Appendix A for snapshots of fibrillisation mixtures). In addition to the faster growth of double-fragment NAC 71-82 peptides, we also observed that fibrils, in agreement with the growth of elongated full-length α-synuclein fibrils, formed at the liquid–air interface [19]. After the growth of an initial population of peptide fibril structures at the liquid–air interface, a sequential growth of other fibril types and morphologies occurred at different container positions. These aggregates were observed to be translucent and trapped in self-supported hydrogels, in line with that predicted by MD data (AP-values). Altogether, the faster aggregation kinetics demonstrated by double-fragment peptides suggested a proximity effect that governs both the rate of fibrillisation and the fibril structure heterogeneity in these peptide systems.

To quantify the extent of fibrillisation and to detect the amount of peptides in a soluble oligomeric state, we calculated the fibrillar yield after 72 h. After the soluble fractions had been isolated from the respective pellet of P2-L(G,G) and of P2-L(G,G)-CT fibrillar samples, we determined the total fibrillar yield to be ~91% and ~77%, respectively. These data were compared to the fibrillar yield of the single-peptide systems P1 and P1-CT, which were ~88% and ~67%, respectively. The higher yield of fibrils in double-fragment peptide systems compared to single-fragment could be explained by the proximity of NAC 71-82 fragments of the double-fragment peptides. In comparison with CD analysis data of the soluble fractions of the P1 fibrillar supernatant [20], the supernatants of the P2-L(G,G) and P2-L(G,G)-CT fibrillar mixtures did not reveal any β-sheet content (Appendix A), despite detectable total peptide content in the supernatant as shown by the BCA assay. 

### 2.3. Raman Spectroscopy Revealed β-Sheet Structure in Double-Fragment NAC 71-82 Peptide Fibrils

To characterise the fibril structure population generated from our designed NAC 71-82 peptides, we used LTRS that allows for trapping and vibrational characterisation of single fibrillar aggregates in an aqueous solution [21,22]. When selecting aggregates to trap and analyse, we selected those in the size range of 2–20 µm. The acquired Raman spectra of the aggregates all displayed a symmetric Amide-I band centred at 1670–1672 cm^−1^, indicating a low random coil and α-helix content, as this would show up as additional peaks around 1680 and 1655 cm^−1^, respectively (for examples of averaged Raman spectra, see Appendix A). To retrieve more information about the secondary structure elements, we performed a peak deconvolution of the Amide-I band. We deconvoluted the Amide-I band into five peaks, located at ~1670, 1655, 1625, 1560 and 1530 cm^−1^ (Figure 2A). Peaks 1670 and 1655 cm^−1^ we assigned to the secondary structures β-sheets and α-helix, respectively [23]. The peak centred on ~1625 cm^−1^ has previously been associated with bound water [24], disordered structures or vibronic coupling [25] and 3-10 Helix [26]. Finally, the peaks at 1560 cm^−1^ and 1530 cm^−1^ are not part of the Amide-I band and were included only to assist in getting an accurate deconvolution, as the area of these peaks overlaps with those in the Amide-I band.

The result of the deconvolution shows that there is a higher abundance of β-sheet structure in the P1 compared to the P1-CT, P2-L(G,G) and P2-L(G,G)-CT systems ~70% versus 40–60% (Figure 2B). We attribute this to the amyloidogenic nature of P1 fibrils when compared to amorphous fibrils prepared from P1-CT peptide, as previously demonstrated [15]. Further, we observed that the α-helix abundance in all sampled fibrillar aggregates was around ~10–30% (Figure 2B), with the P1 system containing the least amount. Both of these results suggest that the P1 fibrils are more mature as a conversion of α-helix to β-sheets has been linked to the maturity of fibrils, as shown by Apetri and co-workers [27].

### 2.4. TEM Data Showed the Presence of Elongated Fibrils in Double-Fragment NAC 71-82 Peptide Fibrils

To analyse the morphology of fibrils, we used transmission electron microscopy (TEM). TEM images show that double-fragment fibrils incubated for 72 h resulted in spheroidal oligomers, shorter protofibrils, protofilaments and more mature fibrils (Figure 3A,B, with representative images in Appendix A). A previous assessment of fibrils from the capped single-fragment (P1) peptide indicated amyloid structure but did not demonstrate fibrils with morphologies similar to that for the full-length protein [15]. Notably, the use of designed double-fragment peptide models resulted in fibrils and morphologies similar to that of the full-length protein [28]. In general, elongated and mature fibrils of the P2-L(G,G) peptide exhibited rod-like structures of ~100–1000 nm in length and either 5 nm (objects measured 1, 2 and 3 in image panel A) or 10 nm (objects 4, 5 and 6 in image panel A) in width (Appendix A, for fibril widths). Fibrils with a width of 5 nm were associated with single protofilaments, and fibrils with a 10 nm width were associated with more mature fibril structures.

In contrast, fibrils generated from the P2-L(G,G)-CT peptide were ~100–500 nm in length and showed a homogenous population of fibrillar widths of 10 nm (objects 1–6 in image panel B). A fraction of the fibrils displayed twisted morphology as shown by torsion points on individual fibrils (Figure 3B). The morphological differences observed for the P2-L(G,G) and P2-L(G,G)-CT fibrils, which were generated in the absence and presence of either physiological salt solution or N- and C-terminal capping, thereby mimicking the influence of pH on total peptide charge, are in line with those reported for in vitro generated full-length α-synuclein fibrils [29]. In contrast, fibrils from single-fragment peptides yielded only shorter protofibrils in bundles (see Appendix A for more TEM examples). It is noteworthy that the P2-L(G,G) peptide system also contained fibrils with similar morphology to that observed in the P1 peptide system. This indicates that the destabilised β-hairpin-like fold of the P2-L(G,G) peptide leads to a destabilised fibril growth, in agreement with MD simulation data.

### 2.5. TCSPC Spectroscopy Can Detect Structural Differences in Double-Fragment NAC 71-82 Peptide Fibrils

Although Raman spectroscopic investigations successfully could detect β-sheet content in fibril mixtures prepared from double-fragment peptides, the structural difference between mature fibrils could not be detected. To be able to address this challenge, we used TCSPC spectroscopy in combination with ThT dye which is used to detect β-sheet structure [30]. In contrast to steady-state fluorescence intensity measurements, which only provide an average of all fluorescent states present in a sample, the use of time-resolved fluorescence spectroscopic techniques aids in resolving fluorescence lifetimes and their relative contribution to the total emission. Previous studies on the use of ThT as a probe for detecting β-sheet structure have suggested that the fluorescence intensity increase upon fibril binding is due to a hindering in intra-ring rotation [31]. This bound state of ThT has an increased quantum yield and demonstrates a longer fluorescence lifetime compared to unbound dye molecules. Previous studies on the binding of ThT to full-length α-synuclein fibrils suggested a bound state with a fluorescence lifetime of ~2 ns [17]. As ThT is a promiscuous molecule and known to bind to both soluble (oligomers and protofibrils) and insoluble (protofilaments and mature fibrils) molecules, it can be suggested that a measurement of the ThT fluorescence decay profile from a complex fibrillar mixture can provide a fingerprint of the morphological features of such a mixture. Measured ThT fluorescence lifetime decay profiles after being incubated with generated fibril samples confirmed the presence of free and bound forms of ThT (Figure 4 and Table 1). Measurements of single-fragment peptide systems, prepared as controls, revealed a ThT fluorescence lifetime of ~0.2 ns (T_1_), in agreement with that found for ThT in pure Tris-HCl buffer, in the absence of peptide. Based on this observation, it could be concluded that this lifetime component is associated with an unbound state of ThT.

The P1 system displayed an additional ThT fluorescence lifetime of ~2 ns (T_2_) with a percentage contribution (amplitude, A_2_) of 21% of the total amount of collected photons. This lifetime was suggested to originate from a bound state of ThT. In contrast, fibrils found in the P1-CT system demonstrated only a single ThT fluorescence lifetime (T_1_), previously ascribed to an unbound state of ThT, thus suggesting an amorphous structure of these fibrils. In relation to fibrils prepared from single-fragment peptides, double-fragment peptide fibrils displayed an increased fraction of bound forms of ThT. This was observed as an increase in the amplitude (A_2_) of the ~2 ns lifetime from 21% (P1) to 33% and 68% for P2-L(G,G) and P2-L(G,G)-CT, respectively. The higher percentage contribution of the longer lifetime component in P2-L(G,G) and P2-L(G,G)-CT systems reflects the higher ThT affinity of these fibrils. It can be suggested that this increased affinity originates from the binding of ThT to mature elongated fibrillar aggregates in these systems. Based on the structural diversity of fibrils observed after TEM analysis, we suggest that the higher percentage contribution of a bound state of ThT in the P2-L(G,G)-CT peptide fibrils could be correlated to the presence of mature elongated fibrils in this sample.

The increase in the fluorescence (T_1_) lifetime—previously associated with a free unbound state of ThT in this peptide system—from 0.2 to 1.0 ns was interpreted as being due to a problem in resolving and separating lifetimes with similar contribution. The 1.0 ns component can be the result of an overlap between the 0.2 ns lifetime component for an unbound state of ThT with a longer lifetime component associated with a second bound state of ThT. In fact, a fitting procedure including three exponentials revealed the presence of a short component of 0.2 ns and two components of 0.6 ns and 1.7 ns; however, the fitting procedure could not resolve their relative contribution.

Overall, this implies that the fibrillar environment experienced by ThT in the P2-L(G,G)-CT system is different than in other investigated fibrillar mixtures. These results shed light on the possibility of using a TCSPC-based diagnostic approach to detect individual fibril conformations from a mixture of fibrils.

## 3. Materials and Methods

### 3.1. Chemicals

The NAC-71-82-based peptide fragments (Table 2) were all purchased as lyophilised trifluoroacetate (TFA) salts from CASLO ApS, Kongens Lyngby, Denmark. All peptide samples were delivered as pre-weighed powders (4.0 mg). Bicinchoninic acid (BCA) analyses were performed using a Pierce^®^ protein assay kit (Thermo Fisher Scientific, Waltham, MA, USA).

Thioflavin T (ThT) (≥65% dye content), NaCl (anhydrous, ≥99%), colloidal silica (LUDOX^®^) particles (40 weight-% suspension in deionised water) and Tris-HCl (≥99.9%, titration) were all purchased from Sigma-Aldrich (St. Louis, MO, USA). Methanol for cleaning cuvettes for circular dichroism (CD) spectroscopy measurements was purchased from VWR Chemicals (Radnor, PA, USA). During preparation of buffers and for preparing capped peptide solutions, Millipore quality (mQ) (Bedford, MA, USA) water was used.

### 3.2. Instruments

Prior to fibrillisation, peptide samples were immersed for 3 min in an ultrasound bath (2510E-DTH, Branson Ultrasonics Corporation, Danbury, CT, USA). Isolation of soluble peptide fractions from harvested fibrillar samples was carried out on an Eppendorf 5424 R benchtop centrifuge (Eppendorf AG, Hamburg, Germany). A Tecan Spark10M multimode plate reader (Tecan Austria GmbH, Grödig, Austria) was employed for the determination of fibrillisation supernatant peptide concentrations (as determined by bicinchoninic acid assay, BCA). Circular dichroism (CD) spectroscopic analysis was executed using a JASCO J-720 Spectropolarimeter (JASCO Corporation, Tokyo, Japan) equipped with a Peltier controller (JASCO PTC-423L, JASCO Corporation, Tokyo, Japan) for temperature regulation. Time-resolved fluorescence spectroscopic measurements were performed using the TCSPC technique on a Fluorolog-3-222 TCSPC spectrometer from HORIBA Jobin Yvon (Kyoto, Japan). This instrument was equipped with a data station hub and a TBX-04 photon detector, both from HORIBA Jobin Yvon, Kyoto, Japan. Transmission electron microscopy (TEM) images were acquired with a Talos L120 TEM (ThermoFisher—FEI, Hillsboro, OR, USA) equipped with a Ceta CMOS 4K × 4K pix (ThermoFisher—FEI, Hillsboro, OR, USA) camera. To trap and acquire Raman spectra on fibrils in solution, we used an optical tweezer instrument. In short, the system was built around an inverted microscope with a water immersion objective (Olympus IX71 equipped with UPlanSApo 60x WIR, Tokyo, Japan). This was to allow for trapping ~50 µm from a surface thus reducing the glass signal. A 100 mW continuous 808 nm laser (Crystalaser DL808-100-S, Reno, NV, USA) was used both as trapping laser and for Raman spectroscopic excitation. Raman measurements were performed using a McPherson 207-789A scanning spectrometer (Chelmsford, MA, USA). To reduce noise, the camera was cooled to −95 °C. The Andor Solis software (v. 4.30, Oxford Instruments, Abingdon, UK) was used to operate the spectrophotometer. A custom-written LabVIEW program (National Instruments, Austin, TX, USA) was used to operate the laser tweezers and the microscope stage. A detailed description of the instrument setup can be found in a previously reported study [22].

### 3.3. Molecular Dynamics Simulations

MD simulations of double-fragment NAC 71-82 peptide aggregation mixtures were performed using a protocol similar to that previously described for single-fragment NAC 71-82 peptides [15]. For a detailed description of the methodology used and the trajectory analyses performed, see the Appendix A “Molecular dynamics (MD) simulation—Peptide aggregation protocol”.

### 3.4. Preparation of Fibrils

Peptide solvation and in vitro fibril preparation were carried out using a protocol previously established in our laboratory [15]. Soluble peptide fractions were isolated from aggregated preparations and isolated as follows. Buffer solution (20 mM Tris-HCl pH 7.3 and 0.15 M NaCl) was added to the pre-weighed lyophilised samples of the P1-CT and P2-L(G,G)-CT peptides. Deionised Millipore quality (mQ) water instead of saline buffer was, due to the high hydrophobicity and limited solubility in isotonic solvents of the capped peptides, added to the P1 and P2-L(G,G) peptide samples. The final peptide concentration in each preparation was 2 mg mL^−1^. Prior to fibrillisation, all samples were treated using repeated steps of vortexing and ultrasound sonication. The peptide preparations were then incubated at 37 °C for 72 h in quartz Suprasil^®^ cuvettes (3.0-mL and 1-cm path length, Hellma GmbH Müllheim, Germany) on an M22/1 magnetic stirrer (Framo Morat GmbH, Eisenbach, Germany) at 1300 rpm. After 72 h, fibril samples were transferred to 2.0 mL plastic tubes (Eppendorf, Hamburg, Germany) at 25 °C and centrifuged at 16,900× *g* for 10 min. The resulting supernatant and pellet were separated and stored at −20 °C.

### 3.5. Bicinchoninic Acid (BCA) Assay

Measurements of the total peptide concentration in soluble fractions recovered from the in vitro fibrillisation of the single (P1 and P1-CT) and double (P2-L(G,G) and P2-L(G,G)-CT) NAC 71-82 peptide preparations were performed using the Pierce^®^ protein assay kit as previously described [20].

### 3.6. Circular Dichroism (CD) Spectroscopy

Soluble fractions recovered from the in vitro fibrillisation samples of single- (P1 and P1-CT) and double-fragment (P2-L(G,G) and P2-L(G,G)-CT) NAC 71-82 peptide preparations were investigated using CD spectroscopy at 25 °C for secondary-structure formation. Fibrillar supernatants (200 μL) of 200 μM (0.25 mg⋅mL^−1^) were measured in total 10 times, and the resulting spectra were presented after the solvent background had been removed. The signal in mdeg was converted into mean residue ellipticity.

### 3.7. Raman Spectroscopy

Crude fibrillar samples (~5 μL) were added to a microscope slide after which a cover slip was applied. Due to the high concentration of fibrils found in the P1 sample, ~1 μL of this sample was diluted 100 times in mQ water prior to analysis. An optical tweezer was thereafter used to trap each fibrillar particle in a light beam, and each particle was thereafter analysed by Raman spectroscopy at 25 °C. In total, 10 fibrillar particles (2–20 μm) were trapped for each peptide system, and their corresponding Raman spectra were recorded, especially analysing the Amide-I vibrational band (1600–1700 cm^−1^). To deconvolute the Amide-I band, we first pre-processed the spectra by subtracting an experimentally determined background and then using asymmetric least-squares baseline correction to remove any residual background [32], an intensity normalisation to the range 0–1 and Savitzky–Golay filtering of width 5 [33]. Then we used the non-linear optimisation toolbox in MATLAB (MathWorks cooperation, Natick, MA, USA [34] to deconvolute the spectra. Specifically, we used an interior-point method to solve a constrained least-squares problem. We assumed that interesting peaks in the Amide-I and -II bands were represented by five Lorentzian peaks with centre positions 1670, 1655, 1625, 1560 and 1540 cm^−1^. We constrained the centre position of each peak to ±10 cm^−1^ of their starting value during optimisation while keeping height and width unconstrained. To obtain their relative abundance, each peak was integrated, and its corresponding area was normalised to each of the total areas of the Amide-I and -II peaks investigated.

### 3.8. Transmission Electron Microscopy (TEM)

Fibrillar samples (3.5 µL) were drop-coated onto carbon-coated copper grid formvar that had been pre-cleaned using glow discharge with PELCO easiGlow (TedPella Inc., Redding, CA, USA). The samples were then negatively stained using 1.5% uranyl acetate. The width and length of selected and representative fibrils observed in captured TEM images were measured using the Image J software (v. 1.52q, National Institute of Health, Bethesda, MD, USA) [35]. Separate TEM images were assembled into panels using the GIMP image program (v. 2.10.8, GIMP Development Team, Berkeley, CA, USA) [36] with no further image processing.

### 3.9. Time-Correlated Single Photon Counting (TCSPC) Spectroscopy

Fibril samples prepared after incubation for 72 h at 37 °C were mixed with a constant concentration of ThT in a total volume of 2 mL in a cuvette (Hellma, 3 mL, 1 cm path length) under continuous stirring at room temperature. The final peptide and ThT concentrations were 0.1 mg⋅mL^−1^ and 10 μM, respectively. The excitation source was a NanoLED (HORIBA Scientific, Kyoto, Japan) generating 455 nm excitation pulses at a 1 MHz repetition pulse. The pulse width of the NanoLED light source was <1.3 ns which enabled a lifetime resolution of ~100 ps. The photon-counting rate was always kept below 2% of the excitation source repetition rate to prevent pile-up effects. Prior to each measurement, the instrument response function (LUDOX solution) was recorded (excitation and emission wavelength both set to 455 nm). Fluorescence decay times of ThT were measured at 485 nm. Emission monochromator bandpasses were set to either 3 nm or 6 nm. The collection of photons stopped when a total number of 10,000 photons were collected in one channel (with a calibration time of ~28 ps/channel). The generated decay profiles were typically fitted using either a single or double exponential decay model within the DAS6 software (HORIBA, Jobin Yvon, UK). Curve fits were generally accepted when χ^2^ < 1.2.

## 4. Conclusions

Here, we designed a series of synthetic peptides based on the NAC 71-82 amino acid fragment of α-synuclein to yield a variety of fibril structures. We characterised the fibrillisation process (CD spectroscopic and BCA analyses) and found, using LTRS, that the β-sheet content in endpoint fibrils was lower in double-fragment compared to single-fragment peptide fibrils. TEM data revealed structural heterogeneity in double-fragment peptide fibrils and, in addition to short protofibril structures in bundles, more elongated fibrils. Elongated fibrils of the non-capped double-fragment peptide were twisted and wider in contrast to fibrils of the capped counterpart which were more rod-like. This suggests that the introduction of capped termini in the double-fragment peptide results in a destabilisation of fibrillar core contacts, in line with the MD simulation data. Fibrils prepared from the non-capped single-fragment peptide and incubated with ThT revealed a single ThT fluorescence lifetime. This lifetime could be ascribed to an unbound state of ThT and confirmed the amorphous nature of these fibrils. In contrast, fibrils prepared from the capped counterpart and both designed double-fragment peptides yielded an additional longer ThT fluorescence lifetime component. The higher percentage contribution of the longer ThT fluorescence lifetime in non-capped compared to capped double-fragment peptide fibrillar samples was believed to be due to the binding of ThT to more elongated and mature fibrils. The results presented in this work show the potential of using peptide models for the generation of fibrils that share morphological features relevant for disease, e.g., twisted and rod-like polymorphs. Moreover, the potential of the TCSPC method for detecting the presence of mature fibrils in a sample is highlighted.

## Figures and Tables

**Figure 1 ijms-22-09334-f001:**
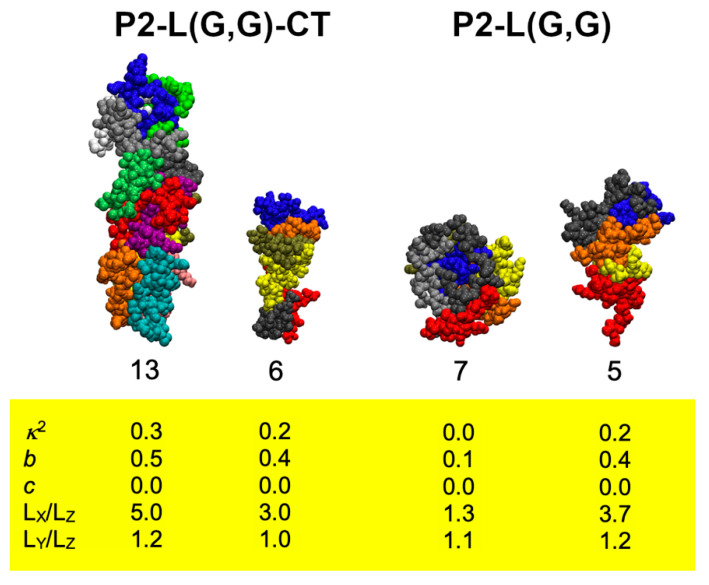
Selected ordered peptide aggregates formed in the P2-L(G,G)-CT (**left**) and P2-L(G,G) (**right**) peptide mixtures studied by MD simulation. Individual peptides in each aggregate (the total number of peptides presented below each complex) are here depicted as coloured van der Waals volumes. Yellow box contains calculated values for the shape descriptors investigated (relative shape anisotropy (κ^2^), asphericity (*b*), acylindricity (*c*) and the ratios of gyration tensors in the *x*, *y* and z dimensions (*L_X,Y,Z_*)).

**Figure 2 ijms-22-09334-f002:**
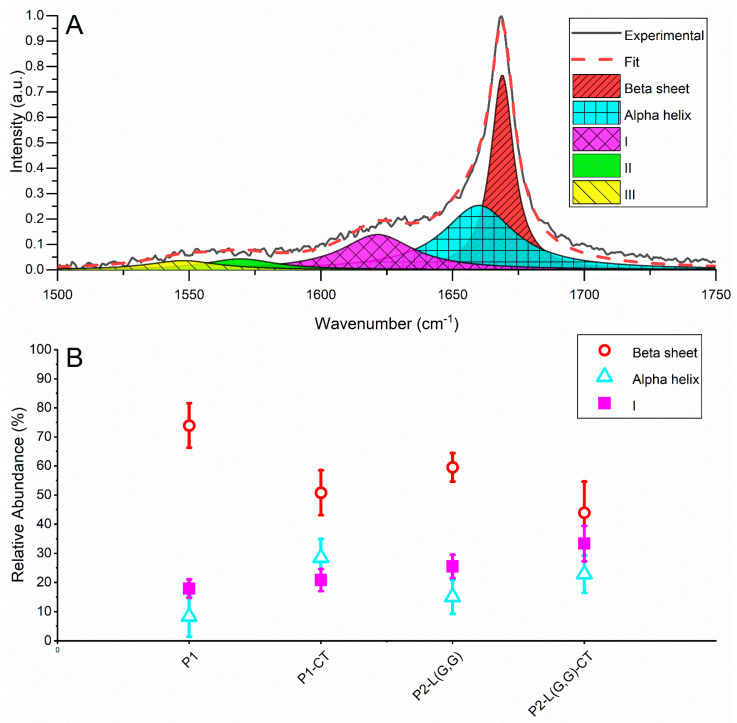
(**A**) A deconvolution example of the Amide-I band showing peak assignment and placement. (**B**) The relative abundance of beta-sheets estimated from the area of the 1670 cm^−1^ peak for the different systems and the relative abundance of α-helix estimated from the area of the 1655 cm^−1^ peak for the different systems.

**Figure 3 ijms-22-09334-f003:**
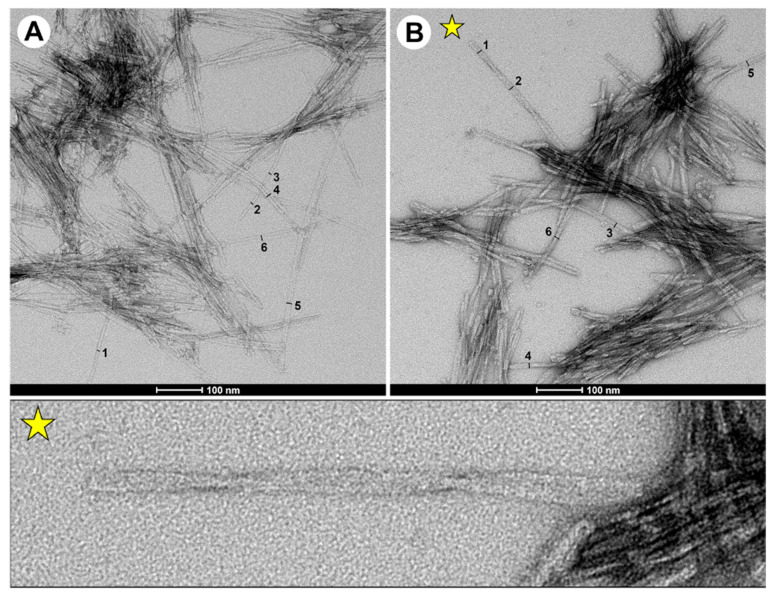
Representative TEM images of (**A**) P2-L(G,G) and (**B**) P2-L(G,G)-CT fibrils generated after 72 h of incubation in mQ water and 20 mM Tris-HCl buffer with 0.15 M NaCl at pH 7.3. In these panels, the numbers (1–6) shown highlight fibrils whose diameters were measured. A yellow star is inserted close to a fibril with a twisted morphology in panel B. A zoomed-in picture of this twisted polymorph with evident torsion points is added for clarity. Scale bars (when shown) are 100 nm.

**Figure 4 ijms-22-09334-f004:**
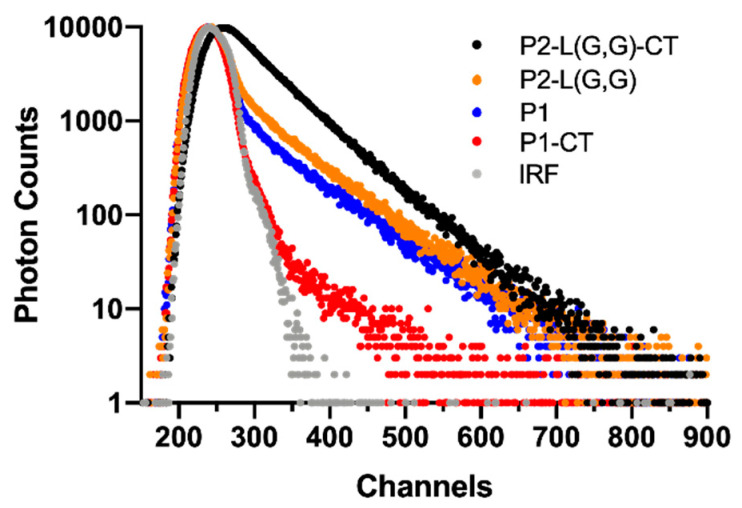
TCSPC analysis of ThT fluorescence of samples comprised of 0.1 mg⋅mL^−1^ of peptide and 10 μM of ThT. Decay profiles are shown for the instrument response function (IRF, grey), P1-CT (red), P1 (blue), P2-L(G,G) (orange) and P2-L(G,G)-CT (black). Each channel was calibrated to 28 ps. The time to reach 10,000 photons in the peak channel was found to be dependent on the relative contribution of short- and long-lived photons. Under the conditions employed, a measurement typically required 3–7 min of photon sampling (longest time for the non-amyloid P1-CT sample).

**Table 1 ijms-22-09334-t001:** TCSPC data from adding 10 uM of ThT to 0.1 mg/mL of fibrils, prepared during continuous stirring for 72 h at 37 °C.

	Lifetime/ns		Amplitudes/%		
Fibril Sample	T_1_	T_2_	A_1_	A_2_	χ^2^
P1-CT	0.189 ± 0.002		100		2.11
P1	0.196 ± 0.003	2.190 ± 0.008	79	21	0.94
P2-L(G,G)	0.223 ± 0.003	2.031 ± 0.006	67	33	0.91
P2-L(G,G)-CT	0.991 ± 0.025	1.816 ± 0.006	32	68	1.15

**Table 2 ijms-22-09334-t002:** Peptides studied and their respective purities (in %) as described by the supplier.

ID ^a^	Amino Acid Sequence ^b^	Purity/%
P1	Ac-VTGVTAVAQKTV-NMe	≥95.3
P1-CT	VTGVTAVAQKTV	≥96.3
P2-L(G,G)	Ac-VTGVTAVAQKTV**GG**VTGVTAVAQKTV-NH_2_	≥97.3
P2-L(G,G)-CT	VTGVTAVAQKTV**GG**VTGVTAVAQKTV	≥95.2

^a^ CT stands for charged termini; ^b^ Ac- and -NMe stand for acetylated N-terminus and methyl-amidated C-terminus, respectively. Bold and underlined amino acids mark the position of the included linker region in the double-fragment (P2) designed peptides.

## Data Availability

All the data are shown in the main paper or in the Appendix A.

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
