# Peer review of "Synthetic NAC 71-82 Peptides Designed to Produce Fibrils with Different Protofilament Interface Contacts"

_ijms, 2021, doi:10.3390/ijms22179334_

Round 1

Reviewer 1 Report

In this manuscript, the authors have described the generation of multiple peptides based on the NAC sequence of a-synuclein for the characterisation of a variety of protofilaments. Through a variety of biophysical techniques, they confirm the b-sheet structure of these protofilaments, while using TCSPC as a means of distinguishing finer features of these protofilament structures, such as the presence of twisted polymorphs.

The experiments are quite scientifically robust, and the observations from authors are well substantiated with orthogonal approaches. However, from the intentions of this manuscript in claiming that TCSPC is a powerful technique that can resolve microstructural differences between protofilaments (from title especially), I have yet to find the current results definitive in this, and would thus like to see more substantial experiments/analysis to support this statement, before I would recommend this manuscript for publication. 

Major issues:

  •  The TCSPC data is interesting; it shows quite clearly that there is significant difference in the photo decay depending on the sample itself. However, without orthogonal approaches to understand the molecular basis of this, I'm not quite sure if one is able to say that these differences observed are due to the individual fibril differences. In particular, due to the fact that ThT is afterall a promiscuous binder in detecting all different amyloid forms, I'm not sure if heterogeneity of a sample can be captured by ThT. At the same time, rather than just microstructural differences, I wonder if macroscopic fibril differences can also result in a difference in TCSPC data (i.e. fibril packing/stacking/bundling leading to less accessibility of ThT, rather than the individual morphological differences).  In that aspect, the authors have showed through their images that altering physiological strength (H2O vs Tris buffer) results in morphological differences. Have the authors also tried TCSPC measurements on fibrils that have been generated in different conditions? With more data compiled on a wider range of fibrils produced (particularly for a single peptide that can generate two different kinds of fibrils in different buffer conditions), the TSCSPC analysis can be more convincing to show in terms of microstructural differences.

Minor issues:

  • Fig S10: I'm not sure if the authors interpretations in the caption describing "flakes" and "stick-like structures" are suitable for the brightfield microscope pictures with the resolution of 20um. The TEM data is much more convincing with their higher resolution as well. I don't actually see why Fig S10 is necessary, and would propose authors remove it so as to avoid confusion and misunderstanding. 
  • Table S2: Title label should be width, not length. It would also be helpful though to include the length measurements in this table that the authors have described in the manuscript.
  • Figure referencing: as the authors have already done so dilligently for the referencing for the supplementary figures, could they also do the same for the main figures? It's just much easier to understand the interpretation of a sentence when there is a specific figure reference to refer to.

Author Response

REVIEWER #1

 Comment 1: The TCSPC data is interesting; it shows quite clearly that there is significant difference in the photo decay depending on the sample itself. However, without orthogonal approaches to understand the molecular basis of this, I'm not quite sure if one is able to say that these differences observed are due to the individual fibril differences. In particular, due to the fact that ThT is after all a promiscuous binder in detecting all different amyloid forms, I'm not sure if heterogeneity of a sample can be captured by ThT. At the same time, rather than just microstructural differences, I wonder if macroscopic fibril differences can also result in a difference in TCSPC data (i.e. fibril packing/stacking/bundling leading to less accessibility of ThT, rather than the individual morphological differences).  In that aspect, the authors have showed through their images that altering physiological strength (H2O vs Tris buffer) results in morphological differences.

Author reply 1: We thank the reviewer for insightful comments regarding his/her concern about the risk of overinterpreting the TCSPC data. We agree with the reviewer that since ThT is known to bind to a variety of beta-sheet structures present in a fibrillisation mixture (oligomers, protofibril, protofilaments and mature fibrils) this fact limits the possibility of assigning the ThT decay profile to a specific fibrillar structure.  In addition to the comment stated by this reviewer (and comments raised by reviewer 2) we have revised the manuscript to better reflect our use of TCSPC for obtaining a morphological “fingerprint” for all fibrillar mixtures analysed and correlate these results to data obtained with the other biochemical and biophysical techniques used in this work.

To further strengthen our conclusions and to provide a more careful interpretation of the TCSPC data several changes have now been introduced in the manuscript:

  • Major parts of the “Abstract” section has been rewritten, starting at line 14
  • The sentence starting at line 123 has been revised from:

“Although Raman spectroscopic techniques have successfully provided information on the extent of the secondary structure of fibrillar aggregates, this technique has so far reached limited success in detecting and separating fibril polymorphs. To circumvent this limit, time-correlated single photon counting (TCSPC) can be used.”

To:

“Although Raman spectroscopic techniques have successfully provided information on the extent of the secondary structure of fibrillar aggregates, this technique has so far reached limited success in detecting fibril polymorphs in crude fibrillar samples. To circumvent this limit, time-resolved fluorescence spectroscopy can be suggested to complement vibrational spectroscopic methods.”

  • The sentence starting at line 144 has been revised from:

“Results from the analyses suggested that this technique can be used to map the presence of different individual fibrillar structures in a complex mixture. Altogether, the results presented in this work show the application of the TCSPC method for detecting the presence of mature fibrils in a sample, as well as to detect finer changes in fibril structures such as those observed between polymorphs in neurodegenerative disease.

To:

“Results from the analyses suggested that this technique can be used to map the presence of mature fibrils in a complex fibrillar mixture. Altogether, the results presented in this work show the potential of using peptide models for the generation of fibrils that share morphological features of relevant for disease e.g., twisted and rod-like polymorphs.”

  • The sentence starting at line 367 has now been changed from:

“Based on the structural diversity of fibrils observed after TEM analysis, we suggest that the higher percentage contribution of a bound state of ThT in the P2-L(G,G)-CT peptide fibrils, could be used to detect finer changes in fibril structure such as e.g. the presence of twisted polymorphs” ­

To:

“Based on the structural diversity of fibrils observed after TEM analysis, we suggest that the higher percentage contribution of a bound state of ThT in the P2-L(G,G)-CT peptide fibrils, could be correlated to the presence of mature elongated fibrils in this sample” ­

  • The sentence starting at line 544 has now been changed from:

“The higher percentage contribution of the longer ThT fluorescence lifetime in non-capped compared to capped double-fragment peptide fibrillar samples was believed to be due to the binding of ThT to more elongated and twisted polymorphs.”

To:

“The higher percentage contribution of the longer ThT fluorescence lifetime in non-capped compared to capped double-fragment peptide fibrillar samples was believed to be due to the binding of ThT to more elongated and mature fibrils.”

  • The sentence starting at line 547 has now been changed from:

“These results suggest the use of TCSPC to detect mature fibrils in a sample and finer changes in fibril structures such as those between rod-like and twisted polymorphs”

To:

“The results presented in this work show the potential of using peptide models for the generation of fibrils that share morphological features of relevant for disease e.g. twisted and rod-like polymorphs. Moreover, the potential of the TCSPC method for detecting the presence of mature fibrils in a sample is highlighted.”

Comment 2: Have the authors also tried TCSPC measurements on fibrils that have been generated in different conditions? With more data compiled on a wider range of fibrils produced (particularly for a single peptide that can generate two different kinds of fibrils in different buffer conditions), the TSCSPC analysis can be more convincing to show in terms of microstructural differences.

Author reply 2: No, we have not analysed a wider population of fibrils than those presented in the current work. Although we agree with the reviewer that TCSPC measurements on more fibril structures that are prepared at different solvent conditions would be interesting and would strengthen our conclusions even further, we however believe that such studies would be outside the scope of the present study.

Comment #3: Fig S10: I'm not sure if the authors interpretations in the caption describing "flakes" and "stick-like structures" are suitable for the brightfield microscope pictures with the resolution of 20um. The TEM data is much more convincing with their higher resolution as well. I don't actually see why Fig S10 is necessary, and would propose authors remove it so as to avoid confusion and misunderstanding. 

Author reply 3: We thank the reviewer for the valuable comment on how to improve our paper and has now removed Figure S10 and all citations to this figure from the main text and the supporting information section (line 260). After removing Figure S10 all other supplementary figures have been renumbered. Moreover, as a consequence of removing Figure S10 we have now also clarified the description of the size population of fibril particles analysed by LTRS, starting at line 487.

Comment 4: Table S2: Title label should be width, not length. It would also be helpful though to include the length measurements in this table that the authors have described in the manuscript.

Author reply 4: We have now changed the Title label in Table S2 according to the reviewer’s comment. Due to stacking and bundling of fibrils the length of fibrils was an estimate based on single measurements on fibrils not only found in Figure 3. Based on this only fibril widths will be reported in Table S2.

Comment 5: Figure referencing: as the authors have already done so dilligently for the referencing for the supplementary figures, could they also do the same for the main figures? It's just much easier to understand the interpretation of a sentence when there is a specific figure reference to refer to.

Author reply 5: We do not fully understand what the reviewer wants us to do. Maybe the reviewer is referring to how we cite figures and tables in the Supporting Information Section and wants us to use a similar strategy also for figures and tables in the main paper? The reason why we added more detailed information when citing Figures and Tables that belong to the Supporting Information Section was to help the reader to follow the story more easily. We believe that this is not necessary for figures and tables belonging to the main text since we are afraid that the readability will suffer from such treatment.

Reviewer 2 Report

The authors present an interesting study on peptides which form fibrils with different protofilament interface contacts. These designed and synthesized peptides are supposed to act as diagnostic tools in a-synucleinopathies. The MD supported and carefully performed design and identification of suitable peptides was the basis for an intensive characterization of the fibril formation process of the investigated species. In this respect the authors made use of several experimental techniques, like Raman-, fluorescence- and CD spectroscopy, TEM. Finally, the authors worked out the potential of using fluorescence life-time analysis. Although the work seems to be solid and give some interesting results, some points need to be addressed before publication.    

Major points:

The authors claim, that TCSP measurements can be a promising tool to obtain more detailed information (e.g. polymorphism) about the fibrils. In my opinion this statement needs some more justification: (i) The authors use ThT for which typically the intensity increase is measured to follow the fibril formation. The authors should also analyze the intensity profile of ThT during the fibrilization and compare this to their life-time information. What do they learn from life-times in addition to that of intensity measurements?

The authors should try to explain why the life-time becomes larger upon fibril formation while the intensity typically increase. Is a process like dye-quenching involved? It would be helpful to have a molecular understanding about how/why the life-time changes appear, in order to rationalize the proposed approach.

In my opinion the TCSP-approach is only a rather small contribution to the total work which is presented here. Therefore, I am not sure whether this aspect should be mentioned explicitly in the title of the paper (in particular if the mechanism is not explained and validated to a certain degree).  

Minor points:

Page 11, Section 4.9: The authors should give the pulse width of the light source.

Page 11, Section 4.9: The authors should give the information how long it typically takes to accumulate 10 000 photons (this would be of interest to judge whether the method is of practical use and/or is superior to intensity measurements).

Author Response

REVIEWER 2

Comment 1: The authors claim, that TCSP measurements can be a promising tool to obtain more detailed information (e.g. polymorphism) about the fibrils. In my opinion this statement needs some more justification: (i) The authors use ThT for which typically the intensity increase is measured to follow the fibril formation. The authors should also analyze the intensity profile of ThT during the fibrilization and compare this to their life-time information.

Author reply 1: If we understand this question correctly, the reviewer asks for an experiment where the fluorescence intensity of ThT is measured in parallel to TCSPC measurements of ThT fluorescence lifetimes during fibrillisation. Although it would indeed be interesting to be able to capture fluorescence intensity changes and lifetime component contributions over the fibrillisation process, this has not been the focus of the current work. We performed our TCSPC measurements after adding ThT to pre-formed fibrils that had been incubated for 72h according to a previous protocol (Näsström et al. Sci. Rep. 2019, 9, 15949-15962) with the sole aim to investigate the possibility of using the TCSPC method to obtain a fluorescence-lifetime-based “fingerprint” for a complex fibril mixture.

Comment 2: What do they learn from life-times in addition to that of intensity measurements?

Author reply 2: Although the observation that the ThT steady-state fluorescence intensity increases upon binding to fibrils, this kind of measurement only provides an average of all fluorescent states present in solution. A complementary technique that provides more detailed information is measurements based on time-resolved fluorescence spectroscopy by e.g. TCSPC. This technique reveals upon collection of single photons at a static emission wavelength the presence and the relative contribution to the total intensity of all dominant fluorescent ThT states in solution. For a molecular probe as ThT in the presence of fibrils these different states mean different bound and unbound states.

It should however be noted that we clearly could observe an increase in the photon count rate upon adding ThT to a fibril mixture which contained beta-sheet rich structures (as observed by other biophysical and biochemical techniques), thus supporting the fact that the ThT fluorescence intensity typically increases during fibril binding in fluorescence steady-state measurements.

Starting at line 335, we have now added more text that clarifies the benefits of using time-resolved fluorescence in addition to steady-state measurements.

“In contrast to steady-state fluorescence intensity measurements, which will only provide an average of all fluorescent states present in a sample, the use of time-resolved fluorescence spectroscopic technique will aid in resolving fluorescence lifetimes and their relative contribution to the total emission”

 Comment 3: The authors should try to explain why the life-time becomes larger upon fibril formation while the intensity typically increase. Is a process like dye-quenching involved? It would be helpful to have a molecular understanding about how/why the life-time changes appear, in order to rationalize the proposed approach.

Author reply 3: Since a quenching process always decreases and not increases fluorescence intensity we are a bit puzzled by the question raised by the reviewer. If the question is about the correlation between ThT fluorescence intensity increase and fluorescence lifetime upon fibril binding, this can be addressed by a change in the conformational flexibility of ThT upon fibril binding.

Since the fluorescence intensity of ThT is increasing upon binding to fibrils this implies that a new state of ThT is formed with a higher quantum yield and hence longer fluorescence lifetime. A known explanation to this behavior is that the ThT molecules that bind to fibrils typically get stabilized in a conformation that hinders inter-ring rotation which results in that fluorescence radiation becomes a more populated deactivation route (see e.g. Malmos K.G. et al. Amyloid 2017, 24, 1-16 and Kitts C.C. et al. J. Phys. Chem. B 2009, 113, 12090-12095). This population of ThT molecules will have a higher quantum yield and a longer fluorescence lifetime than those ThT molecules that are not stabilized by fibril binding. Typically, ThT molecules that are not stabilised by fibril binding will have a low quantum yield and a very fast fluorescence lifetime (0.1ns). ThT dye molecules that are stabilised in the beta-sheet rich structural motif provided by fibrils will be stabilised and therefore demonstrate a higher quantum yield and a longer fluorescence lifetime. Binding of ThT to the full-length alpha-synuclein has previously been reported to be ~2ns (Sulatskaya A.I. et al. Int. J. Mol. Sci. 2018, 19, 2486-2502) in agreement with that for the binding of ThT to NAC 71-82 peptide fibrils. In general, longer fluorescence lifetimes can be correlated to more stabilised fluorophores.

What is interesting is that if more additional fluorescent states of ThT can be observed than those discussed so far, this can imply the presence of more bound states. In this paper we suggest that these different bound states of ThT observed in the presence of  P2-L(G,G)-CT peptide fibrils (revealing a third lifetime of 0.6 ns) can indicate the presence of additional fluorescence states of ThT i.e. ThT binding with different conformations to the fibril structure surface. These additional binding modes of ThT can then be used to identify unique fibril structural motifs.

More text and references (Malmos K.G. et al. Amyloid 2017, 24, 1-16 and Sulatskaya A.I. Int. J. Mol. Sci. 2018, 19, 2486-2502) have now been added starting at line 338 to provide a more complete description on role molecular basis for ThT fluorescence emission and lifetimes:

“Previous studies on the use of ThT as a probe for detecting beta-sheet structure have suggested that the fluorescence intensity increase upon fibril binding is due to a hindering in intra-ring rotation [31]. This bound state of ThT has an increased quantum yield and demonstrate a longer fluorescence lifetime compared to unbound dye molecules. Previous studies on the binding of ThT to full-length alpha-synuclein fibrils suggested a bound state with a fluorescence lifetime of ~2ns [17]. As ThT is a promiscuous molecule and known to bind to both soluble (oligomers and protofibrils) and insoluble (protofilaments and mature fibrils) it can be suggested that a measurement of the ThT fluorescence decay profile from a complex fibrillar mixture can provide a “fingerprint” of the morphological features of such a mixture”

Comment 4: In my opinion the TCSP-approach is only a rather small contribution to the total work which is presented here. Therefore, I am not sure whether this aspect should be mentioned explicitly in the title of the paper (in particular if the mechanism is not explained and validated to a certain degree).  

Author reply 4: We thank the referee for the insightful comments on the role of TCSPC measurements for its impact on the findings presented in this work. We have decided to follow the suggestion made by the reviewer and removed the part in the title describing the TCSPC measurements.

Moreover, the “Abstract” section of the paper, starting at line 14, has now been rewritten to better reflect the change in the title of the manuscript.

Comment 5: Page 11, Section 4.9: The authors should give the pulse width of the light source.

Author reply 5: The pulse width of the Nano-LED light source used in the TCSPC experiments have now been inserted in the Materials and Methods section of the paper starting at line 511:

The pulse width of the Nano-LED light source was < 1.3 ns which enabled a lifetime resolution of ~100ps

Comment 6: Page 11, Section 4.9: The authors should give the information how long it typically takes to accumulate 10 000 photons (this would be of interest to judge whether the method is of practical use and/or is superior to intensity measurements).

Author reply 6: We thank the reviewer for the valuable comment. We have now inserted the following text in the caption of Figure 4, starting at line 389:

“The time to reach 10 000 photons in the peak channel was found to be dependent on the relative contributions of short and long-lived photons. Under the conditions employed, a measurement typically required 3-7 min of photon sampling (longest time for the non-amyloid P1-CT sample).”

Round 2

Reviewer 1 Report

I appreciate the authors' efforts in revising the manuscript, which I believe has improved the manuscript considerably. They have addressed my concerns, notably the interpretation of the TCSPC data, and thus I would recommend this manuscript suitable for publication.

Reviewer 2 Report

Can now be published